# G Protein-Coupled Receptor Kinase 2 Selectively Enhances β-Arrestin Recruitment to the D_2_ Dopamine Receptor through Mechanisms That Are Independent of Receptor Phosphorylation

**DOI:** 10.3390/biom13101552

**Published:** 2023-10-20

**Authors:** Marta Sánchez-Soto, Noelia M. Boldizsar, Kayla A. Schardien, Nora S. Madaras, Blair K. A. Willette, Laura R. Inbody, Christopher Dasaro, Amy E. Moritz, Julia Drube, Raphael S. Haider, R. Benjamin Free, Carsten Hoffman, David R. Sibley

**Affiliations:** 1Molecular Neuropharmacology Section, National Institute of Neurological Disorders and Stroke, National Institutes of Health, 35 Convent Drive, Bethesda, MD 20892, USAfreeb@mail.nih.gov (R.B.F.); 2Institut für Molekulare Zellbiologie, CMB-Center for Molecular Biomedicine, Universitätsklinikum Jena, Friedrich-Schiller-Universität Jena, Hans-Knöll-Straße 2, D-07745 Jena, Germanyraphael.haider@med.uni-jena.de (R.S.H.); carsten.hoffmann@med.uni-jena.de (C.H.); 3Division of Physiology, Pharmacology and Neuroscience, School of Life Sciences, Queen’s Medical Centre, University of Nottingham, Nottingham NG7 2UH, UK; 4Centre of Membrane Protein and Receptors, Universities of Birmingham and Nottingham, Birmingham B15 2TT, UK

**Keywords:** GRK, D_2_ receptor, phosphorylation, β-arrestin

## Abstract

The D2 dopamine receptor (D2R) signals through both G proteins and β-arrestins to regulate important physiological processes, such as movement, reward circuitry, emotion, and cognition. β-arrestins are believed to interact with G protein-coupled receptors (GPCRs) at the phosphorylated C-terminal tail or intracellular loops. GPCR kinases (GRKs) are the primary drivers of GPCR phosphorylation, and for many receptors, receptor phosphorylation is indispensable for β-arrestin recruitment. However, GRK-mediated receptor phosphorylation is not required for β-arrestin recruitment to the D2R, and the role of GRKs in D2R–β-arrestin interactions remains largely unexplored. In this study, we used GRK knockout cells engineered using CRISPR-Cas9 technology to determine the extent to which β-arrestin recruitment to the D2R is GRK-dependent. Genetic elimination of all GRK expression decreased, but did not eliminate, agonist-stimulated β-arrestin recruitment to the D2R or its subsequent internalization. However, these processes were rescued upon the re-introduction of various GRK isoforms in the cells with GRK2/3 also enhancing dopamine potency. Further, treatment with compound 101, a pharmacological inhibitor of GRK2/3 isoforms, decreased β-arrestin recruitment and receptor internalization, highlighting the importance of this GRK subfamily for D2R–β-arrestin interactions. These results were recapitulated using a phosphorylation-deficient D2R mutant, emphasizing that GRKs can enhance β-arrestin recruitment and activation independently of receptor phosphorylation.

## 1. Introduction

Dopamine receptors (DARs) are members of the GPCR super-family and consist of five structurally distinct subtypes [1,2,3]. These can be divided into two subfamilies on the basis of their structure, function, and pharmacological properties [3]. The “D_1_-like” subfamily consists of the D_1_ and D_5_ receptors (D1R and D5R), which couple to the heterotrimeric G proteins G_S_ or G_olf_ to stimulate adenylyl cyclase activity and raise intracellular levels of cAMP. The D_2_-like subfamily includes the D_2_, D_3_, and D_4_ receptors, which couple to inhibitory G_i/o/z_ proteins to reduce adenylyl cyclase activity as well as modulate voltage-gated K^+^ or Ca^2+^ channels. Within the central nervous system, these receptors modulate movement, learning and memory, reward and addiction, cognition, and certain neuroendocrine functions. DARs are also critically important drug targets. D_1_-like agonists are used to regulate blood pressure, while D2R/D3R agonists are used to treat the symptoms of Parkinson’s disease. Further, all FDA-approved antipsychotics function as D2R antagonists. As with other GPCRs, DARs are subject to a wide variety of regulatory mechanisms, which can either positively or negatively modulate their expression or functional activity.

One common form of GPCR regulation is that of agonist-induced desensitization, which is typically initiated via phosphorylation of the activated GPCR by one or more members of the G protein-coupled receptor kinase family (GRKs). GRKs are serine/threonine protein kinases and consist of seven isoforms that can be divided into three subfamilies based on their structure [4,5,6]. The visual GRKs, GRK1 and GRK7, are restricted to the retina and modulate rod and cone opsins, respectively. GRK2 and GRK3 represent a second subfamily and are ubiquitously expressed. GRK4/5/6 constitute a third subfamily and are also widely distributed, with the exception of GRK4, which is expressed in only a few tissues. Once phosphorylated by GRKs, GPCRs recruit arrestin proteins that serve to terminate G protein-mediated signaling and also to initiate receptor internalization through clathrin-coated pits [4,5,6]. It is now clear that interaction with an activated GPCR is typically necessary for the activation of β-arrestins. This can occur through multiple mechanisms, including interactions of β-arrestin with the cytoplasmic surface of the receptor core and/or with phosphorylated regions of the receptor, such as its carboxyl terminus or cytoplasmic loops [7,8,9,10,11]. Once activated, β-arrestins can engage with clathrin-coated endocytic structures or initiate downstream G protein-independent signaling events [9,10,12,13].

While some DAR subtypes, such as the D1R [14,15], follow the canonical model of GPCR regulation as described above, the D2R appears to be an exception to this rule. Using site-directed mutagenesis and in situ phosphorylation assays, we previously identified all of the GRK-mediated phosphorylation sites on D2R and, using phosphorylation-deficient receptor mutants, demonstrated that phosphorylation does not regulate D2R interactions with β-arrestin [16,17]. Rather, we [17] and others [18,19] have shown that GRK-mediated phosphorylation of the D2R mediates its post-endocytic trafficking, specifically by enhancing its rate of recycling to the cell surface. Nonetheless, overexpression of GRKs has been shown to enhance β-arrestin recruitment to the D2R in various heterologous expression systems [16,17,20,21,22], although the specific mechanisms remain unclear. Notably, however, we [17] and others [21] have shown that D2R-G protein coupling and activation is not required for GRK2 interactions with the D2R or for the modulation of its activity.

Similarly, the GRK specificity for regulating D2R signaling is not entirely clear. Most studies have implicated the GRK2/3 subfamily in modulating the D2R [17,21,22,23,24,25], although it is not known if GRK2 and GRK3 are equally effective in this regard. Further, the role of other GRKs, such as GRK6, in regulating the D2R has been postulated [26,27]. We have now used HEK293 cell lines in which the expression of specific GRK isoforms were selectively eliminated using CRISPR/Cas9 technology [28] to address the GRK specificity question and to further explore how GRKs regulate β-arrestin recruitment to the D2R and trigger internalization. We confirm that GRK2/3 are predominately, but not exclusively, involved in modulating β-arrestin recruitment to the D2R, with GRK2 seeming to be the most critical isoform in HEK293 cells. However, agonist-activated D2Rs can still recruit β-arrestin even in the absence of GRKs or receptor phosphorylation, suggesting that the modulation of β-arrestin recruitment to the D2R can be achieved through multiple mechanisms.

## 2. Materials and Methods

### 2.1. Materials

Cell culture media and reagents were purchased from Invitrogen (Carlsbad, CA, USA) and Corning (Glendale, AZ, USA). Cell culture flasks and all assay plates were purchased from ThermoFisher Scientific (Waltham, MA, USA) and Greiner Bio-One (Monroe, NC, USA). Dopamine was obtained from Sigma-Aldrich (St. Louis, MO, USA). [^3^H]-Sulpiride was purchased from Perkin Elmer Life Sciences (Waltham, MA, USA) or Novandi (Södertälje, Sweden). Compound 101 was purchased from Tocris Bioscience (Minneapolis, MN, USA). HEK293 cells originated from the American Type Culture Collection (Manassas, VA, USA). HEK293 cell lines in which the expression of individual or multiple GRK isoforms was eliminated using CRISPR/Cas9 technology were generated as described in Drube et al., 2022 [28]. Plasmid constructs expressing Gα_o1_-Rluc8, Gβ1, Gγ2-mVenus, CAMYEL biosensor, β-arrestin2-mVenus, and human D2R-Rluc8 were kind gifts from Drs. Jonathan Javitch and Hideaki Yano at Columbia University (New York, NY, USA). Plasmid constructs expressing GRK2-rGFP, LYN-rGFP, and β-arrestin2-Rluc2 were kind gifts from Dr. Michel Bouvier at the University of Montreal (Montreal, QC, Canada). Plasmid constructs for GRK2, GRK3, GRK5, and GRK6 were kind gifts from Dr. Jeffrey L. Benovic at Thomas Jefferson University (Philadelphia, PA, USA). Plasmid constructs consisting of Nanoluciferase fused to the C-terminus of β-arrestin2 along with FlAsH biosensors incorporated into different positions in β-arrestin2 (Nluc-β-arrestin2-FlAsH biosensors) were generated as previously described [29]. The plasmid expressing the phosphorylation-null (PO_4_-null) D2R was generated as described previously [17]. Experiments that directly compared WT and PO_4_-null D2R used rat D2RL (long isoform). Human D2RL (long isoform) was used in all other experiments. MiniComplete™ protease inhibitor mixture was purchased from Millipore Sigma (Burlington, MA, USA). Anti-GRK2 antibodies were from Santa Cruz Biotechnology (Dallas, TX, USA, sc-13143, mouse). Anti-GAPDH antibodies were from Proteintech (Rosemont, IL, USA, 10494-1-ap, rabbit). Horseradish peroxidase-conjugated anti-rabbit-IgG and anti-mouse-IgG were from Jackson ImmunoResearch Laboratories (West Grove, PA, USA, #111-035-003, #115-035-003).

### 2.2. Cell Culture and Transfections

HEK293 cells were cultured in Dulbecco’s modified Eagle’s medium (DMEM) with 4.5 g/L glucose, L-glutamine without sodium pyruvate supplemented with 10% fetal bovine serum, 100 U/mL penicillin, and 100 μg/mL streptomycin. Cells were grown at 37 °C in 5% CO_2_ with 90% humidity. Cells seeded in 100 mm or 35 mm plates were transfected with DNA constructs using PEI (1 µg DNA:3 µL PEI). Complete cell culture media were replaced with non-supplemented DMEM 10 min prior to transfection. Non-supplemented media were replaced with complete media the following day.

### 2.3. β-Arrestin2 Recruitment BRET Assay

HEK293 cells were transiently transfected with the indicated D2R-Rluc8 construct (35 mm plate: 0.1 μg; 100 mm plate: 0.5 μg), β-arrestin2-mVenus (35 mm plate: 1 μg; 100 mm plate: 5 μg), and the indicated GRK construct (35 mm plate: 1 μg; 100 mm plate: 5 μg) or pcDNA control (empty vector). Cells were washed and resuspended in Dulbecco’s phosphate-buffered saline (DPBS) with 200 µM sodium metabisulphite (to prevent dopamine from oxidizing) and 5.5 µM glucose. Cells were then plated in 96-well white solid bottom plates (Greiner, Monroe, NC, USA) and incubated at room temperature for 45 min. To determine concentration response curves, cells were incubated with 5 µM coelenterazine h (Nanolight Technology, Pinetop, AZ, USA) for 5 min and then stimulated with the indicated concentrations of dopamine for 10 min. For kinetic experiments, the cells were incubated with 5 µM coelenterazine h for 5 min, then stimulated with either vehicle or 100 µM dopamine followed by BRET assessment over time. BRET signals were determined by calculating the ratio of the fluorescence intensity emitted by mVenus (535/30 nm) over the luminescence emitted by Rluc8 (475/30 nm) using a Pherastar plate reader (BMG Labtech, Cary, NC, USA).

### 2.4. LYN BRET Receptor Internalization Assay

HEK293 cells were transiently transfected with the indicated D2R-Rluc8 (35 mm plate: 0.1 μg; 100 mm plate: 0.5 μg) and LYN-rGFP (35 mm plate: 1 μg; 100 mm plate: 5 μg) constructs and the indicated GRK construct (35 mm plate: 1 μg; 100 mm plate: 5 μg) or pcDNA control. Cells were washed and resuspended in Dulbecco’s phosphate-buffered saline (DPBS) with 200 µM sodium metabisulphite and 5.5 µM glucose. Cells were then plated in 96-well white solid bottom plates (Greiner, Monroe, NC, USA) and incubated at room temperature for 45 min. Cells were then incubated with the indicated concentrations of dopamine for 25 min followed by incubation with 2 µM Prolume Purple (Nanolight Technology, Pinetop, AZ, USA) for 5 min. BRET signals were determined by calculating the ratio of the fluorescence intensity emitted by rGFP (515 nm) over the luminescence emitted by Rluc8 (410 nm) using a Pherastar plate reader (BMG Labtech, Cary, NC, USA).

### 2.5. G_o_ Heterotrimer Activation BRET Assay

HEK293 cells were transiently transfected with the Gα_o1_-Rluc8 (35 mm plate: 0.1 μg; 100 mm plate: 0.5 μg), Gβ_1_ (35 mm plate: 0.9 μg; 100 mm plate: 4.5 μg), and Gγ_2_-mVenus (35 mm plate: 1 μg; 100 mm plate: 5 μg) constructs and the indicated untagged D2R construct (35 mm plate: 1 μg; 100 mm plate: 5 μg). Cells were washed and resuspended in Dulbecco’s phosphate-buffered saline (DPBS) with 200 µM sodium metabisulphite and 5.5 µM glucose. Cells were then plated in 96-well white solid bottom plates (Greiner, Monroe, NC, USA) and incubated at room temperature for 45 min. Cells were incubated with 5 µM coelenterazine h (Nanolight Technology, Pinetop, AZ, USA) for 5 min and then stimulated with the indicated concentrations of dopamine for 5 min. BRET signals were determined by calculating the ratio of the fluorescence intensity emitted by mVenus (535/30 nm) over the luminescence emitted by Rluc8 (475/30 nm) using a Pherastar plate reader (BMG Labtech, Cary, NC, USA).

### 2.6. cAMP CAMYEL BRET Assay

HEK293 cells were transiently transfected with the indicated untagged D2R construct (35 mm plate: 1 μg; 100 mm plate: 5 μg) and the CAMYEL cAMP biosensor (yellow fluorescence protein-Epac-Rluc) (35 mm plate: 1 μg; 100 mm plate: 5 μg) [30]. Cells were washed and resuspended in Dulbecco’s phosphate-buffered saline (DPBS) with 200 µM sodium metabisulphite and 5.5 µM glucose. Cells were then plated in 96-well white solid bottom plates (Greiner, Monroe, NC, USA) and incubated at room temperature for 45 min. Cells were incubated with 5 µM coelenterazine h (Nanolight Technology, Pinetop, AZ, UAS) for 5 min and then stimulated with 10 µM forskolin to increase intracellular cAMP levels along with the indicated concentrations of dopamine for 10 min. BRET signals were determined by calculating the ratio of the fluorescence intensity emitted by mVenus (535/30 nm) over the luminescence emitted by Rluc8 (475/30 nm) using a Pherastar plate reader (BMG Labtech, Cary, NC, USA).

### 2.7. GRK2-GFP Recruitment BRET Assay

HEK293 cells were transiently transfected with the indicated D2R-Rluc8 (35 mm plate: 0.1 μg; 100 mm plate: 0.5 μg) and GRK2-rGFP (35 mm plate: 1 μg; 100 mm plate: 5 μg) constructs. Cells were washed and resuspended in Dulbecco’s phosphate-buffered saline (DPBS) with 200 µM sodium metabisulphite and 5.5 µM glucose. Cells were then plated in 96-well white solid bottom plates (Greiner, Monroe, NC, USA) and incubated at room temperature for 45 min. Cells were incubated with 2 µM Prolume Purple (Nanolight Technology, Pinetop, AZ, USA) for 5 min and then stimulated with the indicated concentrations of dopamine for 5 min. BRET signals were determined by calculating the ratio of the fluorescence intensity emitted by mVenus (515 nm) over the luminescence emitted by Rluc8 (410 nm) using a Pherastar plate reader (BMG Labtech, Cary, NC, USA).

### 2.8. Intramolecular β-Arrestin2 FlAsH Assays

HEK293 cells were transiently transfected with the indicated D2R construct (35 mm plate: 0.12 μg), the indicated Nluc-β-arrestin2-FlAsH biosensor (35 mm plate: 1.2 μg), and either pcDNA or the GRK2 construct (35 mm plate: 1 μg). A total of 48 h after transfection, cells were harvested, washed with DPBS, and incubated for 30 min with 250 nM FlAsH in labeling buffer (12.5 µM ethane-1.2-dithiol (EDT), 150 mM NaCl, 10 mM HEPES, 25 mM KCl, 4 mM CaCl_2_, 2 mM MgCl_2_, and 10 mM glucose) as described previously [29,31]. During this incubation, cells were mixed every 5 min via tube inversion. To reduce nonspecific labeling, cells were spun down and resuspended twice and then incubated in 250 µM EDT for 10 min. Cells were then resuspended in 140 mM NaCl, 10 mM HEPES, 5.4 mM KCl, 2 mM CaCl_2_, and 1 mM MgCl_2_ and plated in 96-well white solid bottom plates (Greiner, Monroe, NC, USA). Finally, cells were incubated for 15 min with 100 µM dopamine or vehicle control (200 µM sodium metabisulphite) in the presence of furimazine (N1572 Promega, Madison, WI, USA, 1:7700 ratio). BRET signal was determined by calculating the ratio of the light emitted by FlAsH (535/30 nm) over that emitted by Nluc (475/30 nm) using a Pherastar plate reader (BMG Labtech, Cary, NC, USA). Ligand-induced net BRET was calculated by subtracting the BRET ratio of vehicle-treated cells from the BRET ratio of dopamine-treated cells. This value was then divided by the BRET ratio of vehicle-treated cells and then multiplied by 100.

### 2.9. [^3^H]-Sulpiride Intact Cell Binding Assays

Cell surface D2R expression was determined as described previously [16] using intact cell assays and the membrane impermeant radioligand [^3^H]sulpiride. Cells were seeded into poly-D-lysine-coated 6-well plates 1 day before the assay at a density of 0.6 × 10^6^ cells/well. A total of 24 h after plating, cells were incubated in the presence of either 200 µM sodium metabisulfite (control) or 200 µM sodium metabisulfite plus 30 μM dopamine in DMEM for 1.5 h at 37 °C. Stimulation was terminated by rapidly cooling the plates on ice and washing the cells three times with ice-cold Earle’s Balanced Salt Solution (EBSS). Cells were then incubated with 0.5 mL of [^3^H]sulpiride in EBSS (final concentration, 4.5 nM) at 4 °C for 3.5 h. Nonspecific binding was determined in the presence of 7.5 μM (+)-butaclamol. Cells were washed three times with ice-cold EBSS and removed from plates with 0.5 mL of 1% Triton X-100 and 5 mM EDTA in EBSS. Samples were mixed with 2 mL of liquid scintillation mixture and counted with a Beckman LS6500 scintillation counter. Cells used to measure protein concentration were incubated with EBSS without [^3^H]sulpiride, and a BCA assay was performed to determine total cellular protein concentration per well.

### 2.10. Western Blotting

HEK293 cells seeded in 35 mm dishes were transiently transfected with WT D2R-Rluc8 (0.5 µg), β-arrestin2-mvenus (5 µg), and the indicated amount of GRK2 construct or pcDNA control. A total of 48 h after transfection, the cells were washed in ice-cold EBSS supplemented with calcium and magnesium, then lysed in ice-cold RIPA buffer supplemented with protease inhibitors (cOmplete, Sigma, St. Loius, MO, USA). The samples were rotated at 4 °C for 1 h and the supernatant was isolated following centrifugation at 20,000× *g* at 4 °C for 20 min. Cell lysates were resolved with 4–12% SDS-PAGE and transferred to PVDF membrane. Blots were blocked with 5% milk in TBST (1× Tris-buffered saline and 0.1% Tween 20) for 1 h at room temperature. For detection of GRK2 protein, the blots were incubated with anti-GRK2 antibody or anti-GAPDH antibody overnight at 4 °C at 1:500 or 1:10,000 dilution, respectively. Membranes were washed with TBST and then incubated with secondary antibodies coupled to horseradish peroxidase at 1:10,000 dilution in TBST with 5% milk for 1 h at room temperature. Membranes were washed, and immunoreactive proteins were visualized using chemiluminescence (SuperSignal West Pico PLUS chemiluminescent substrate, (Thermo Fisher Scientific, Waltham, MA, USA) using a ChemiDoc MP Imaging System (Bio-Rad, Hercules, CA, USA).

### 2.11. Data Analysis and Statistics

Data were analyzed using GraphPad Prism 9 (GraphPad Software, San Diego, CA, USA). Concentration-response curves were fit to a log(agonist) vs. response with three parameters and transformed by subtracting baseline response. Data are presented graphically as the mean ± SEM values derived from at least three individual experiments each performed in either triplicate or duplicate, as indicated. EC_50_ and Emax values were calculated from individual concentration-response curves and then averaged to generate mean ± SEM values. Statistical tests were performed using GraphPad Prism as described in the figure legends.

## 3. Results

### 3.1. GRKs Play an Essential, Isoform-Specific Role in Modulating Interactions between β-Arrestin2 and the D2R

To investigate the roles that specific GRK isoforms play in modulating D2R–β-arrestin2 interactions, we initially overexpressed individual GRK isoforms in HEK293 cells and examined dopamine-stimulated β-arrestin2 recruitment to the D2R (Figure 1, Table 1). We focused on GRK2, GRK3, GRK5, and GRK6 as they are ubiquitously expressed and represent the two major subfamilies of non-visual GRKs. β-arrestin2 recruitment was measured using the bioluminescence resonance energy transfer (BRET) assay depicted schematically in Figure 1A. In HEK293 cells expressing endogenous levels of GRKs, dopamine stimulation promoted robust recruitment of β-arrestin2 to the D2R (Figure 1A). We next observed that overexpressing GRK2, GRK3, GRK5, or GRK6 enhanced dopamine-stimulated β-arrestin2 recruitment to the D2R (Figure 1A). While the overexpression of each GRK isoform resulted in a potentiation of β-arrestin2 recruitment, the nuances of these effects were isoform-specific. Whereas the overexpression of all GRK isoforms significantly increased the maximum response to dopamine (Emax), only GRK2 and GRK3 overexpression increased the potency (decrease in the EC_50_) of dopamine for stimulating β-arrestin2 recruitment (Figure 1A, Table 1).

Importantly, our β-arrestin2 BRET recruitment assay only detects physical interactions between the D2R and β-arrestin2, not the functional responses resulting from this interaction. We thus examined D2R internalization as, upon agonist activation of numerous GPCRs including the D2R [32,33,34], β-arrestin2 will scaffold with the AP2/clathrin complex to promote receptor endocytosis. To measure D2R internalization, we used the LYN BRET assay depicted schematically in Figure 1B. This biosensor measures the decrease in BRET between the D2R and a plasma membrane-anchored protein (Lyn kinase) upon receptor internalization [35]. As with the β-arrestin2 recruitment results, dopamine stimulation promotes receptor internalization, and the overexpression of all GRK isoforms potentiates the efficacy (Emax) of the dopamine response (Figure 1B, Table 1). Further, the overexpression of GRK2 or GRK3, but not GRK5 or GRK6, promoted a dramatic increase in the potency (up to 55-fold) of dopamine for stimulating D2R internalization (Table 1). Taken together, these results suggest that while both subfamilies of GRKs are capable of potentiating D2R–β-arrestin2 interactions, and the subsequent receptor internalization, only the GRK2/3 subfamily can increase the potency of dopamine for promoting these effects.

As HEK293 cells endogenously express GRK2, GRK3, GRK5, and GRK6 [28], the interpretation of GRK overexpression experiments in the presence of endogenous GRKs, as described above, can be difficult. Thus, to investigate the role of GRK isoforms in modulating D2R function in more detail, we made use of isoform-specific GRK knockout (KO) cells [28]. These HEK293 cell lines lack either a single GRK isoform (GRK2 KO, GRK3 KO, GRK5 KO, GRK6 KO), the isoforms of a given GRK subfamily (GRK2/3 KO, GRK 5/6 KO), or all four GRKs endogenously expressed in HEK293 cells (ΔQ-GRK KO). Importantly, we initially determined that D2R-G protein-mediated signaling was not significantly affected in any of these GRK KO cell lines using both G_o_ activation and cAMP accumulation assays (Appendix A, Appendix A). We then measured β-arrestin2 recruitment and receptor internalization in each of these cell lines to determine the contribution of each GRK isoform to these processes (Figure 2, Table 2). Notably, in the ΔQ-GRK KO cell line, β-arrestin2 recruitment to the D2R was dramatically reduced, but not completely eliminated (Figure 2A). Interestingly, dopamine-stimulated β-arrestin2 recruitment to the D2R in both the GRK2 KO and GRK2/3 KO cell lines was reduced to nearly the same degree as that seen in the ΔQ-GRK KO cell line (Figure 2A, Table 2). The knockout of any of the other GRK isoforms, GRK3, GRK5, GRK6, or GRK5/6, had no significant effects on dopamine-stimulated β-arrestin2 recruitment (Figure 2A, Table 2). Similarly, we found that dopamine-stimulated D2R internalization was below detectable limits in both the ΔQ-GRK KO and GRK2/3 KO cell lines and was severely reduced in the GRK2 KO cell line (Figure 2B, Table 2). In contrast, there were no significant effects of knocking out the other GRK isoforms on D2R internalization (Figure 2B, Table 2). These results suggest that the GRK2/3 subfamily (and particularly GRK2) is involved in regulating β-arrestin2 recruitment to the D2R and the subsequent internalization in HEK293 cells.

We further examined the GRK isoform specificity for modulating D2R function by expressing single GRK isoforms in the ΔQ-GRK KO cells and assessing β-arrestin2 recruitment and receptor internalization. When transfected into the ΔQ-GRK KO cells, each GRK isoform was capable of rescuing dopamine-stimulated β-arrestin2 recruitment and D2R internalization to a similar degree as that observed in parental HEK293 cells (Figure 2C,D, Table 3). These data demonstrate that all GRK isoforms can promote β-arrestin2 recruitment to the D2R when expressed at sufficient levels. In agreement with our experiments using parental HEK293 cells, the expression of GRK2 or GRK3 in the ΔQ-GRK KO cells promoted a significant increase in dopamine potency (decrease in EC_50_) for promoting both β-arrestin2 recruitment (~10-fold) and D2R internalization (~40-fold), whereas the expression of GRK5 or GRK6 had no effect on the EC_50_ for dopamine (Figure 2C,D, Table 3). These results bolster the notion that GRK2 and GRK3 enhance D2R–β-arrestin2 interactions in a manner that fundamentally differs from that of GRK5 and GRK6. However, an alternative explanation for this observation might be suboptimum levels of GRK5 or GRK6 expression in our assays. To evaluate this possibility, we used GRK5 as a test case and performed a titration of GRK5 plasmid DNA transfected into the ΔQ-GRK KO cells while measuring β-arrestin2 recruitment to the D2R and observed that, even when transfecting maximal amounts of GRK5, only an increase in dopamine efficacy (Emax) was observed with no change in dopamine potency (EC_50_) (Appendix A). These results support the idea that GRK2 or GRK3 can interact in a unique fashion with the D2R to enhance β-arrestin2 interactions.

As the above experiments use genetic approaches to assess GRK isoform specificity, we wished to perform orthogonal pharmacological experiments to assess the role of GRKs in D2R–β-arrestin2 interactions. To do this, we used compound 101 (Cmpd101), a small heterocyclic molecule that binds within the catalytic sites of GRK2 and GRK3 to inhibit their enzymatic activities [36]. Cmpd101 selectively inhibits the GRK2/3 subfamily as it lacks affinity for other GRK isoforms [28,36]. Importantly, using a G_o_ activation BRET assay, we initially determined that Cmpd101 treatment of HEK293 cells does not alter D2R signaling through G proteins (Appendix A). We next evaluated the effects of Cmpd101 on β-arrestin2 recruitment to the D2R (Figure 3A, Table 4). Cmpd101 was found to inhibit dopamine-stimulated β-arrestin2 recruitment to the D2R in a dose-dependent fashion; however, even at maximally effective concentrations (10–100 µM), Cmpd101 was unable to completely inhibit this response. Interestingly, the residual amount of dopamine-stimulated β-arrestin2 recruitment observed in the presence of 100 µM Cmpd101 (Emax ~25% of vehicle control, Table 4) is similar to that seen in the ΔQ-GRK KO cells, which lack GRKs (Emax ~25% of parental HEK293 cells, Table 2). In Figure 3B, we evaluate the effects of Cmpd101 on dopamine-stimulated D2R internalization. Cmpd101 dose-dependently inhibited receptor internalization and was able to completely block this response at a maximally effective concentration of 100 µM (Table 4). Again, these results are comparable to those found with dopamine-stimulated receptor internalization using the ΔQ-GRK KO cells in that the elimination of all GRK expression completely eliminated D2R internalization (Figure 2B, Table 2).

We have previously shown that dopamine can stimulate the physical association of GRK2 with the D2R using a BRET assay [16] and were interested in determining if Cmpd101 could affect GRK2–D2R interactions. Interestingly, we found that Cmpd101 treatment inhibited GRK2–D2R interactions in a dose-dependent fashion (Figure 3C), but was unable to completely block this response, similar to our observations with Cmpd101 in the β-arrestin2 recruitment assay (Figure 3A). Notably, Pack et al., 2018, previously found no effect of compound 101 on D2R–GRK2 interactions using a similar BRET assay [21]. The reason(s) for these discrepant results is not immediately clear but might be due to differences in the physical orientation of the BRET biosensors incorporated into the D2R and GRK2, thus potentially negating the effects of Cmpd101 in the study of Pack et al., 2018 [21]. Importantly, the effects of Cmpd101 on inhibiting GRK2 interactions with the D2R could, at least partially, explain its effects on β-arrestin2 recruitment or on receptor internalization. Taken together, these data further suggest that dopamine-stimulated D2R–β-arrestin2 recruitment and its functional sequelae largely relies on GRK2/3 activity in HEK293 cells.

### 3.2. GRKs Can Enhance β-Arrestin2 Recruitment to the D2R and Receptor Internalization in the Absence of D2R Phosphorylation

Typically, GRK phosphorylation of GPCRs is thought to mediate β-arrestin recruitment, receptor internalization, and downstream signaling pathways [4,5,6,14,37]. Our lab has previously mapped out all the GRK-mediated phosphorylation sites within the D2R [16,17] as well as those mediated by PKCs [38], which together comprise all the sites of known phosphorylation within the D2R when expressed in HEK293 cells. We used this information to create mutant D2Rs that are deficient in either GRK- or PKC-mediated phosphorylation, or both, in which all the GRK and PKC sites are simultaneously mutated to create a completely phosphorylation-deficient D2R (phospho-null D2R) (Appendix A). Further, we previously found that, upon agonist-stimulation, the GRK phosphorylation-null (GRK (-)) D2R was able to recruit β-arrestin to the same extent as the WT D2R [17], suggesting that GRK2 phosphorylation of the D2R is not necessary for β-arrestin recruitment. We wished to extend these findings using the total phospho-null D2R (Appendix A), which is completely incapable of undergoing any agonist-stimulated phosphorylation [17]. Initially, we used a CAMYEL BRET cAMP accumulation assay to show that the phospho-null D2R signals through G proteins in a manner indistinguishable from that of the WT D2R (Appendix A). Next, we compared dopamine-stimulated β-arrestin2 recruitment to the WT and phospho-null D2Rs and found that this response was also indistinguishable between these two receptor constructs (Figure 4A). We also examined dopamine-stimulated GRK2–D2R interactions using a BRET assay and found no differences between the WT D2R and phospho-null D2R in their ability to physically associate with GRK2 upon dopamine stimulation (Figure 4B). These results confirm that, with agonist activation, GRK2 is recruited to the phospho-null D2R; however, phosphorylation of the D2R is not required for its interaction with β-arrestin2.

**Figure 4 biomolecules-13-01552-f004:**
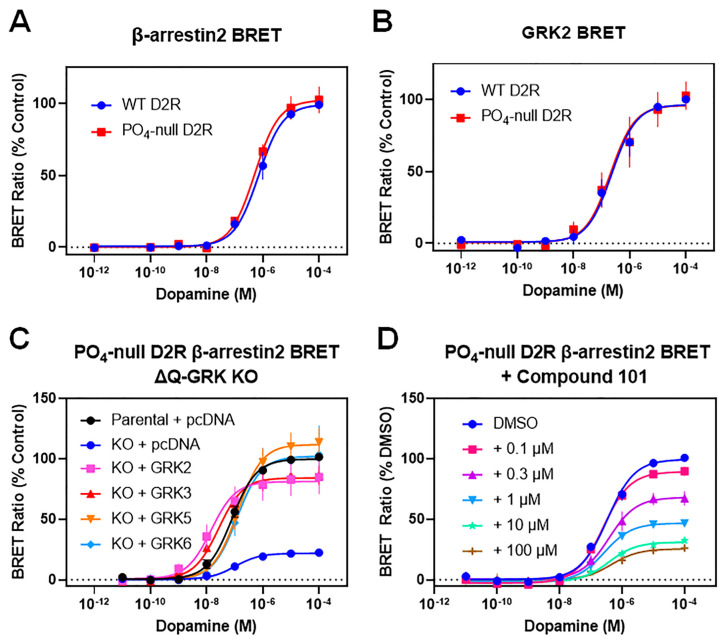
Receptor phosphorylation is not required for GRK2 modulation of dopamine-stimulated recruitment of β-arrestin2 to the D2R. (**A**) HEK293 cells were transfected with either WT or PO_4_-null D2R-Rluc8 constructs along with β-arrestin2-mVenus, and BRET was assessed as described in Figure 1 and Section 2. Data are expressed as a percentage of the maximum dopamine response elicited by the WT D2R and are displayed as mean ± SEM values from at least three experiments yielding the following curve parameters: WT D2R: EC_50_ = 970 ± 460 nM, Emax = 100%; PO_4_-null D2R: EC_50_ = 530 ± 72 nM, Emax = 103 ± 8.9%. Statistical differences between the curve parameters from the WT D2R and PO_4_-null D2R assays were assessed using a *t*-test and found not be significant (*p* > 0.05). (**B**) HEK293 cells were transfected with either WT or PO_4_-null D2R-Rluc8 constructs along with GRK2-rGFP, and BRET was assessed as described in Figure 3C and Section 2. Data are expressed as a percentage of the maximum dopamine response elicited by the WT D2R and are displayed as mean ± SEM values from at least three experiments yielding the following curve parameters: WT D2R: EC_50_ = 530 ± 200 nM, Emax = 100%; PO_4_-null D2R: EC_50_ = 980 ± 590 nM, Emax = 102 ± 10%. Statistical differences between the curve parameters for the WT D2R and PO_4_-null D2R were assessed using a *t*-test and found not to be significant (*p* > 0.05). (**C**) The PO_4_-null D2R and β-arrestin2 were transfected into either parental HEK293 cells or the ΔQ-GRK KO cells. The ΔQ-GRK KO cells were also transfected with either GRK2, GRK3, GRK5, GRK6, or pcDNA (control). Data are expressed as a percentage of the maximum dopamine response observed in parental HEK293 cells and are displayed as mean ± SEM values from at least three independent experiments. Average curve parameters (EC_50_ and Emax values) and statistical comparisons are shown in Table 5. (**D**) HEK293 cells were transfected with PO_4_-null D2R-Rluc8 with β-arrestin2-mVenus, and BRET was assessed in the absence (DMSO) or presence of increasing concentrations of compound 101. Data are expressed as a percentage of the maximum dopamine response observed in cells treated with DMSO and are displayed as mean ± SEM values from at least three independent experiments. Average curve parameters (EC_50_ and Emax values) and statistical comparisons are shown in Table 6.

We next wished to determine if GRK overexpression could enhance β-arrestin2 recruitment to the phospho-null D2R as we previously found with the WT D2R. We chose to use the ΔQ-GRK KO cells for this experiment and found that, as with the WT D2R, the genetic knockout of GRKs significantly reduced β-arrestin2 recruitment to the phospho-null receptor (Figure 4C). Further, the transfection of individual GRKs rescued dopamine-stimulated β-arrestin2 recruitment to levels similar to those observed in parental cells (Figure 4C, Table 5). These results indicate that the GRK-mediated enhancement of β-arrestin2 recruitment is not mediated by receptor phosphorylation. Notably, the GRK-mediated increase in dopamine potency (decrease in EC_50_) that is unique to the GRK2/3 subfamily was also observed using the phospho-null D2R, additionally suggesting that this phenomenon is not mediated by receptor phosphorylation (Figure 4C, Table 5). To directly compare the effects of GRK2 on the WT D2R and phospho-null D2R within the same experiment, we examined dopamine-stimulated β-arrestin2 recruitment using the ΔQ-GRK KO cells (Appendix A). Notably, GRK2 enhanced β-arrestin2 recruitment to the WT D2R and phospho-null D2R in an identical fashion. Finally, we measured β-arrestin2 recruitment to the phospho-null D2R in the presence of Cmpd101 (Figure 4D). As observed with the WT D2R, Cmpd101 dose-dependently decreased β-arrestin2 recruitment to the phospho-null D2R but did not completely eliminate it. Taken together, these results illustrate that GRKs can promote dopamine-stimulated β-arrestin2 recruitment to the D2R through mechanisms that do not involve receptor phosphorylation but apparently involve GRK2 catalytic activity, at least partially.

Having established that receptor phosphorylation is not required for GRK-mediated β-arrestin2 recruitment, we next examined receptor internalization using the phospho-null D2R. We first used intact cell radioligand binding assays that quantitate receptors on the cell surface to examine agonist-induced D2R internalization and found that both the WT and phospho-null D2Rs internalize to the same extent upon stimulation with dopamine (Figure 5A). These results recapitulate those that we previously observed using a related GRK(-) mutant D2R [17]. We next used LYN BRET as an orthogonal assay to assess receptor internalization and found that, following dopamine treatment, the phospho-null D2R internalized to the same extent as the WT D2R (Figure 5B). To assess GRK modulation of the phospho-null D2R internalization, we performed the LYN BRET assay in the ΔQ-GRK KO cells with or without expression of GRKs. In agreement with our results using the WT D2R, the phospho-null D2R loses its ability to internalize in the absence of GRKs (Figure 5C). When individual GRK isoforms are expressed in the ΔQ-GRK KO cells, however, the phospho-null D2R regains its ability to undergo agonist-stimulated internalization (Figure 5C). Further, as was observed with the WT D2R, both GRK2 and GRK3 promote a leftward shift in the dopamine concentration response curve (decrease in EC_50_), whereas GRK5 and GRK6 do not (Figure 5C). These results further confirm that GRKs can promote D2R internalization through mechanisms that do not involve receptor phosphorylation.

It is widely appreciated that β-arrestins undergo distinct conformational changes following their translocation to and interaction with GPCRs [7,8,9,10,11,39]. These conformational changes have been shown to be associated with β-arrestin-mediated cellular responses, such as receptor trafficking and ERK1/2 phosphorylation [9,40]. Consequently, we sought to understand the role of receptor phosphorylation and GRKs in promoting β-arrestin2 conformational changes induced by D2R activation. To study these phenomena, we utilized several of the NanoLuc luciferase (NanoLuc)- and fluoresceine arsenical hairpin binder (FlAsH)-based β-arrestin2 biosensors recently described by Haider et al. (2022) [29]. For each of these biosensors, the FlAsH-binding sites are incorporated into residues at specific locations (numbered 1–5) within the β-arrestin2 protein to report on conformational changes that are structurally specific to the utilized position. Conformational changes in β-arrestin2 following WT D2R or phospho-null D2R activation with or without GRK2 overexpression in HEK293 cells were measured using FlAsH sensors 1–5 and depicted in a radar chart following quantitation (Figure 6A). We found that the conformational changes relative to FlAsH positions 2, 3, and 5 provided the most robust signal following dopamine stimulation and D2R activation. Consequently, data from those FlAsH sensors were transformed into the histograms shown in Figure 6B. Interestingly, following agonist activation, the phospho-null D2R promoted more robust conformational changes in β-arrestin2 than the WT D2R at all three FlAsH positions (Figure 6B). Even more strikingly, overexpression of GRK2 significantly potentiated the phospho-null D2R-mediated β-arrestin2 conformational changes at the FlAsH 3 and FlAsH 5 positions (Figure 6B). Taken together, these data show that D2R phosphorylation is not required for agonist stimulation to promote conformational changes in β-arrestin2 and, further, that GRK2 can potentiate these conformational changes without phosphorylating the receptor. We consequently conclude that GRKs can modulate D2R-mediated β-arrestin2 recruitment and activation through mechanisms that do not involve phosphorylating the receptor.

### 3.3. A Catalytically Inactivating Mutation of GRK2 Reveals Unique Mechanisms of GRK2-Mediated Enhancement of D2R–β-Arrestin2 Interactions

Since phosphorylation of the D2R appears not to be involved in GRK2-mediated enhancement of β-arrestin2 recruitment and activation, we investigated if the GRK2 kinase activity was required for these modulatory effects. To answer this question, we utilized a GRK2 mutant in which an invariant lysine residue at position 220 within the catalytic domain of the protein is changed to an arginine residue (K220R GRK2). This has the effect of rendering GRK2 catalytically inactive, although it can still form complexes with activated GPCRs, which can block the association of endogenous GRKs with receptors, thus functioning as a dominant-negative mutant [41,42]. Because of the latter attribute of this mutant GRK2, we chose to compare the K220R GRK2 with the WT GRK2 using the ΔQ-GRK KO cells, which are devoid of endogenous GRKs. Figure 7A,B show dopamine-stimulated β-arrestin2 recruitment to either the WT D2R or the phospho-null D2R expressed in either parental HEK293 cells or in ΔQ-GRK KO cells that have also been transfected with either WT GRK2 or K220R GRK2 (or pcDNA as a control). The dopamine-stimulated response (Emax) for both the WT and phospho-null D2R in ΔQ-GRK KO cells is only ~20% of that observed in the parental HEK293 cells (Figure 7, Table 7). The expression of WT GRK2 in the ΔQ-GRK KO cells rescued the dopamine-stimulated β-arrestin2 recruitment response for both the WT and phospho-null D2R to levels similar to, if not greater than, those observed in the parental HEK293 cells (Figure 7A,B). Notably, the expression of WT GRK2 in the ΔQ-GRK KO cells shifted the dopamine concentration response curve to the left (decrease in EC_50_) in addition to an increase in the maximum response (Emax) (Table 7). These results with the WT GRK2 recapitulate those shown in Figure 2C and Figure 4C.

Interestingly, the expression of K220R GRK2 in the ΔQ-GRK KO cells also rescued the dopamine-stimulated β-arrestin2 recruitment response for both the WT and phospho-null D2R (Figure 7A,B). However, the expression of the K220R GRK2 only affected the efficacy of the dopamine response (Emax) without an effect on dopamine potency (EC_50_) (Table 7). Thus, the effect of the K220R GRK2 on enhancing β-arrestin2 recruitment to the D2R fundamentally differs from that of the WT GRK2. In fact, the effects of expressing the K220R GRK2 appear more similar to those of expressing GRK5 or GRK6 in the ΔQ-GRK KO cells (Figure 2C and Figure 4C), which only resulted in an increase in the Emax of the dopamine-stimulated β-arrestin2 recruitment response. These results argue that the kinase activity of GRK2 is required to enhance the potency (EC_50_) of dopamine for stimulating β-arrestin2 recruitment to the D2R, but not for enhancing the efficacy (Emax) of this response.

We next wished to investigate the effects of the K220R GRK2 on dopamine-stimulated receptor internalization using the LYN BRET assay. Figure 7C,D show dopamine-stimulated internalization of either the WT D2R or the phospho-null D2R expressed in either parental HEK293 cells or in ΔQ-GRK KO cells that had been transfected with either WT GRK2 or K220R GRK2. As observed previously (Figure 2C and Figure 4C), dopamine-stimulated receptor internalization is essentially non-existent in the ΔQ-GRK KO cells, whereas the expression of the WT GRK2 results in a significant rescue of the dopamine-stimulated internalization response. Notably, this includes an increase in the potency for dopamine using both the WT and phospho-null D2R in comparison with their EC_50_ values observed using the parental HEK293 cells (Table 7). The expression of the K220R GRK2 also rescued the dopamine-stimulated internalization response for both the WT and phospho-null D2R in the ΔQ-GRK KO cells (Figure 7C,D), although there was less of an Emax increase observed with the phospho-null D2R. Notably, however, in contrast to the WT GRK2, expressing the K220R GRK2 in the ΔQ-GRK KO cells had no effect on the potency of dopamine compared with that in the parental cells (Table 7). Again, these receptor internalization results with the K220R GRK2 appear similar to those observed with expressing GRK5 or GRK6 in the ΔQ-GRK KO cells (Figure 2D and Figure 5C). Thus, as with the dopamine-stimulated β-arrestin2 recruitment response, the kinase activity of GRK2 is needed to enhance the potency of dopamine for stimulating internalization of both the WT and phospho-null D2R.

The results in Figure 7 show that both the WT GRK2 and K220R GRK2 can enhance dopamine-stimulated β-arrestin2 recruitment to the D2R, but only the WT GRK2 can increase dopamine’s potency for stimulating this response. While these results imply that the kinase activity of GRK2 is required for the latter effect, we wanted to rule out other possibilities for why the K220R GRK2 seemed defective in this regard. Initially, we examined the kinetics for dopamine-stimulated β-arrestin2 recruitment to the D2R in the absence or presence of either WT GRK2 or K220R GRK2 in ΔQ-GRK KO cells (Appendix A). Interestingly, dopamine-stimulation of β-arrestin2 recruitment appeared to occur more rapidly in the presence of WT GRK2, in contrast to K220R GRK2, but after 20 min these responses appeared to be equivalent, which was the assay time utilized in Figure 7A,B, suggesting that these responses were at equilibrium. We next performed titrations of the WT GRK2 and K220R GRK2 plasmid DNA transfected in the ΔQ-GRK KO cells and measured β-arrestin2 recruitment to the D2R (Appendix A). Surprisingly, for each GRK2 construct, enhancement of the dopamine-stimulated β-arrestin2 recruitment response appears to be at, or near, maximal using the lowest amount of plasmid DNA transfected (100 ng). Further, with all quantities of DNA transfected, the WT GRK2 promoted a shift to the left (increase in potency) in the dopamine concentration-response curve and an increase in Emax (Appendix A), whereas the K220R GRK2 only promoted an increase in Emax (Appendix A). These results suggest that the amount of K220R GRK2 is not limiting in these assays. However, to further investigate this, we expressed both the WT GRK2 and K220R GRK2 in ΔQ-GRK KO cells and performed Western blotting to assess the quantities of GRK2 expressed (Appendix A). Using this approach, we determined that the expression levels of the K220R GRK2 were similar to those of the WT GRK2 using all plasmid transfection quantities, although there was a statistically insignificant trend toward lower expression of K220R GRK using the highest amounts (1000 and 1500 ng) of plasmid DNA (Appendix A). Our conclusion is that the inability of the K220R GRK2 to affect dopamine’s potency for stimulating β-arrestin2 recruitment or receptor internalization is most likely due to its inability to function as a protein kinase.

## 4. Discussion

In the current study, we investigated the mechanisms and specificity of GRK-mediated enhancement of β-arrestin recruitment to the D2R and an associated functional response–receptor internalization. When examined in HEK293 cells, we found that overexpression of GRK2, GRK3, GRK5, and GRK6 all enhanced the efficacy of dopamine for stimulating β-arrestin recruitment to the D2R. In addition, GRK2 and GRK3 promoted a leftward shift in the dopamine concentration-response curve such that the EC_50_ for dopamine was reduced. This phenomenon was also observed for dopamine-stimulated receptor internalization. These initial results led to the following conclusions: (1) when overexpressed in HEK293 cells, all of the tested GRKs were capable of enhancing β-arrestin recruitment as well as D2R internalization, and (2) only GRK2/3 enhanced the potency of dopamine for stimulating these responses. This suggests that: (1) GRK isoform specificity for regulating D2R signaling exists but is “nuanced” in nature and that (2) more than one mechanism exists for enhancing β-arrestin recruitment—one shared by all of the GRK isoforms and another restricted to the GRK2/3 subfamily.

To address GRK specificity in greater detail, we used HEK293 cells in which the endogenous expression of either single GRK isoforms (i.e., GRK2, GRK3, GRK5, or GRK6), both isoforms of a GRK subfamily (i.e., GRK2/3 or GRK5/6), or all GRK isoforms were eliminated using CRISPR-Cas9 technology [28]. It is important to note that GRK2 and GRK6 are expressed at higher levels than GRK3 and GRK5 in the parental HEK293 cells [28,43]. We found that knocking out GRK5 and/or GRK6 expression had no effect on dopamine-stimulated β-arrestin recruitment or receptor internalization. However, knocking out the expression of either GRK2 alone, GRK2 with GRK3, or all four GRKs together attenuated β-arrestin recruitment to about 30–40% of the control Emax response. Taken at face value, these results suggest that at levels of endogenous GRK expression, GRK2 appears to be the most relevant isoform for modulating D2R–β-arrestin interactions. This might be because the expression of GRK2 is considerably higher than that of GRK3 in HEK293 cells [43]. Notably, when overexpressed in the ΔQ-GRK KO cells, all of the GRK isoforms enhanced dopamine-stimulated β-arrestin recruitment and receptor internalization in a similar fashion as when overexpressed in the parental cells. That is, the expression of GRK2 and GRK3 increased the potency and efficacy of dopamine, whereas the expression of GRK5 and GRK6 only enhanced the efficacy of dopamine for stimulating these responses. Thus, when expressed at high levels, GRK3 seems to enhance D2R–β-arrestin interactions and receptor internalization in a manner similar to GRK2. Interestingly, similar results were observed using the µ-opioid receptor in that knocking out GRK2 expression in HEK293 cells exerted a greater effect on attenuating agonist-stimulated β-arrestin recruitment to the µ opioid receptor than knocking out GRK3 [44].

It was interesting to note that in the complete absence of GRK expression, the D2R was still able to recruit β-arrestin when stimulated with dopamine, albeit to only 30–40% of the extent seen in the presence of GRKs. This result suggests that, in the absence of GRKs, the interaction of β-arrestin with the active conformation of the D2R is suboptimal and that GRK expression (GRK2 in particular) is required for effective interactions. Since the D2R–G protein interactions were found to be normal in these cells, these results also indicate that the active state for recruiting β-arrestin differs from that for recruiting G proteins. Evidence for this has also been provided by Pack et al., 2018, using mutant D2Rs that are either G protein or β-arrestin signaling biased [21]. We have also provided evidence that the active states of the D2R differ for recruiting G proteins and β-arrestins using a mutant D2R [45]. Interestingly, in the HEK293 KO cell lines lacking GRK2, we found that dopamine-stimulated receptor internalization was barely detectable or absent, suggesting that either the amount of β-arrestin recruited to the D2R in these cells is insufficient for triggering the internalization process or that, in the absence of GRK2, the D2R is impaired in the *activation* of β-arrestin. Alternatively, GRK2 may exert a facilitatory effect on internalization that does not involve β-arrestin.

We also investigated the isoform specificity of GRK-mediated enhancement of β-arrestin recruitment to the D2R using compound 101, a small molecule that selectively inhibits GRK2 and GRK3. Titration of parental HEK293 cells with compound 101 dose-dependently reduced dopamine-stimulated β-arrestin recruitment to the D2R and receptor internalization, in agreement with the notion that GRK2 enhances these processes. However, similar to the GRK2 KO experiments, maximally effective concentrations of compound 101 were unable to completely inhibit the β-arrestin response, although it completely inhibited receptor internalization. Thus, pharmacological inhibition of endogenous GRK2 mimicked the effects of knocking out GRK2 expression. Even with complete inhibition of GRK2, the agonist-activated D2R is still able to interact with β-arrestin, albeit to a reduced extent. This observation further suggests that either there is a GRK-independent mechanism for D2R-mediated β-arrestin recruitment, or that the functions of GRKs in this response are simply facilitatory as opposed to enabling in nature.

To further investigate the interactions of GRK2 and the D2R, we used a D2R-GRK2 BRET assay that measures the physical proximity of D2R and GRK2. We found that dopamine dose-dependently stimulated a BRET signal between the D2R and GRK2 and that compound 101 dose-dependently inhibited this interaction. Notably, however, maximally effective concentrations of compound 101 were unable to completely inhibit the D2R-GRK2 BRET response, similar to its effects on the β-arrestin recruitment response. Compound 101 is known to bind in the catalytic site of GRK2 to inhibit its enzymatic activity [36,46]; however, to our knowledge, this is the first observation of compound 101 directly inhibiting the interaction of GRK2 with a GPCR. It is conceivable that compound 101 stabilizes GRK2 in a state with lower affinity for GPCRs, although this will need to be directly tested. Clearly, it is possible that the effects of compound 101 on D2R–GRK2 interactions could completely account for its effects on β-arrestin recruitment to the D2R.

GRK modulation of β-arrestin recruitment to GPCRs typically occurs through the phosphorylation of serine and/or threonine residues within cytoplasmic regions of the GPCR, resulting in an increase in affinity for GPCR–β-arrestin interactions, as well as facilitating GPCR-mediated activation of β-arrestins [5,47]. We previously mapped out all of the GRK phosphorylation sites within the D2R and showed that simultaneous mutation of these sites eliminated agonist-stimulated receptor phosphorylation; however, β-arrestin recruitment to the D2R and receptor internalization were unimpaired [16,17]. We have now extended these findings using a mutant D2R in which both the GRK and PKC [38] phosphorylation sites are simultaneously mutated, creating a D2R that is completely null for phosphorylation. Interestingly, using the D2R-GRK2 BRET assay, we found that these mutations did not affect dopamine-stimulated D2R–GRK2 interactions. Thus, when activated by agonists, GRK2 is recruited to the phosphorylation-null D2R, and they physically interact even though receptor phosphorylation does not occur. Interestingly, when the phosphorylation-null D2R was expressed in the ΔQ-GRK KO cells, dopamine-stimulation of β-arrestin recruitment and receptor internalization were impaired in a similar fashion as for the wild-type (WT) D2R. More importantly, when the individual GRKs were expressed with the phosphorylation-null D2R, the agonist-stimulated β-arrestin recruitment and receptor internalization responses were rescued in a similar fashion as for the WT D2R. All GRK isoforms enhanced the efficacy (Emax) of dopamine, whereas only GRK2 and GRK3 enhanced dopamine potency (decreased EC_50_). These results indicate that GRK modulation of the β-arrestin recruitment and receptor internalization responses occur through mechanisms that do not involve D2R phosphorylation.

As GPCRs are intimately involved in the activation of β-arrestins [8,10,29], we used several NanoLuc luciferase (NanoLuc)- and fluoresceine arsenical hairpin binder (FlAsH)-based β-arrestin biosensors, described by Haider et al. (2022), to assess conformational changes associated with β-arrestin activation [29]. WT or phosphorylation-null D2Rs were transfected into the ΔQ-GRK KO cells and dopamine-stimulated β-arrestin activation was measured with BRET. Surprisingly, more robust conformational changes in β-arrestin were observed with the phospho-null D2R compared with the WT D2R, and the overexpression of GRK2 significantly potentiated the phospho-null D2R-mediated β-arrestin conformational changes. As previously noted, we [17] and others [18,19] have shown that GRK-mediated phosphorylation of the D2R mediates its post-endocytic trafficking, specifically by enhancing its rate of recycling to the cell surface. It is tempting to speculate that the enhanced conformational changes in β-arrestin as detected using Nluc-β-arrestin2-FlAsH biosensors 2, 3, and 5 might be involved in directing the intracellular trafficking of the D2R, perhaps through orchestrating its interactions with post-endocytic sorting proteins [19]. Alternatively, the β-arrestin activation state could be involved in biasing the signaling of the D2R [22,29], although these possibilities will need to be tested directly. Taken together, these results further demonstrate that D2R phosphorylation is not required for dopamine stimulation to promote β-arrestin activation and that GRK2 can potentiate this process without phosphorylating the D2R.

Our results raised an obvious question: if receptor phosphorylation is dispensable for D2R-mediated β-arrestin recruitment and activation, then what is the role of GRKs in modulating these responses for the D2R, and is kinase activity even required? To address these questions, we used the K220R kinase-defective mutant of GRK2, which still forms active complexes with GPCRs [41,42]. We transfected the WT or K220R GRK2 constructs along with the WT or phosphorylation-null D2R constructs in the ΔQ-GRK KO cells and then measured dopamine-stimulated β-arrestin recruitment and receptor internalization. To summarize the results, we found that the K220R GRK2 could rescue the dopamine-stimulated β-arrestin recruitment and receptor internalization responses for both the WT and phospho-null D2R; however, expression of the K220R GRK2 only enhanced the efficacy (Emax) of the dopamine responses without an increase in dopamine’s potency (decrease in EC_50_). These results argue that the kinase activity of GRK2 (or GRK3) is required to enhance the potency of dopamine for stimulating β-arrestin recruitment to the D2R but is not required for enhancing the efficacy of this response. It is not immediately clear what GRK2 substrate may be involved in promoting this effect, although we have previously speculated that another regulatory protein may be involved in modulating D2R–β-arrestin interactions [16,17].

Importantly, a clue may come from studies measuring the desensitization of K^+^ currents elicited by the short (D2S) and long (D2L) D2R isoforms in *Xenopus* oocytes [48,49]. Desensitization of both the D2S and D2L receptors was found to be mediated by β-arrestin and GRK2; however, Ca^2+^ was also required for desensitization of the D2S receptor. These results suggest the existence of a Ca^2+^-dependent protein involved in regulating D2R activity, at least for the D2S isoform. However, we have not found any differences between the D2S and D2L isoforms with respect to their phosphorylation by GRK2 or their desensitization using either cAMP or GTPγS binding assays ([16], and unpublished observations). (The D2L isoform was used in the present study). These differences may be due to the cell types utilized (*Xenopus* oocytes vs. HEK293 cells) or the functional readouts (K^+^ currents vs. cAMP/GTPγS binding) that were assessed, although the future interrogation of Ca^2+^-dependent proteins known to interact with GRK2 (see below) might prove to be informative. Finally, it should be noted that we have performed phospho-proteomics scans of the ΔQ-GRK KO cells with or without GRK2 expression in search of novel GRK2 substrates; however, these experiments were inconclusive (unpublished observations).

Interestingly, the effects of the K220R GRK2 mutant in enhancing the efficacy of dopamine-stimulated β-arrestin recruitment to the D2R appeared similar to those of GRK5 and GRK6, neither of which have been shown to phosphorylate the D2R [17]. Taken together, these results suggest that there is a kinase-independent mechanism for GRK2 enhancement of D2R–β-arrestin interactions. While entirely speculative, it is conceivable that, when associated with the D2R, GRKs may provide a non-catalytic scaffolding function, perhaps in association with another protein, or proteins, that enhances β-arrestin recruitment to the receptor. In support of this hypothesis is the observation that GRK2 can interact with and affect the activity of dozens of various proteins, many in a phosphorylation-independent manner [5,50,51,52].

In summary, our results have confirmed and extended our previous findings that receptor phosphorylation is not required for β-arrestin recruitment to, nor activation by, the D2R. Rather, it appears that GRK-mediated phosphorylation of the D2R primarily mediates its post-endocytic trafficking, as previously shown [17,18,19]. We have now provided evidence that, at least in HEK293 cells, GRK2 is the specific GRK isoform for regulating D2R function, and that two different GRK-mediated mechanisms exist for enhancing β-arrestin recruitment to the D2R. One of these involves the phosphorylation of a yet-to-be-determined GRK2 substrate, while the second appears to be independent of GRK catalytic activity and may involve a scaffolding function. Future experiments will be directed at the elucidation of these novel GRK-mediated mechanisms of D2R regulation.

## Figures and Tables

**Figure 1 biomolecules-13-01552-f001:**
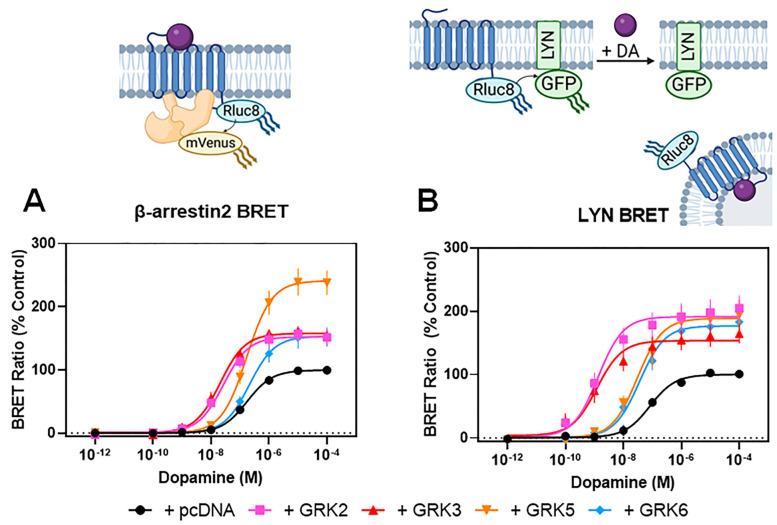
GRK isoforms have differential effects on β-arrestin2 recruitment to the D2R and on receptor internalization. HEK293 cells were transiently transfected with plasmids for D2R-Rluc8 and either GRK2, GRK3, GRK5, GRK6, or pcDNA (control), along with additional plasmids for the assay components as described in Section 2. (**A**) Concentration-response curves for dopamine stimulation of β-arrestin2-mVenus recruitment to D2R-Rluc8 using the β-arrestin2 BRET assay were generated as described in Section 2. (**B**) D2R-Rluc8 and LYN-rGFP were transfected with the indicated GRK isoforms followed by measurement of dopamine-stimulated receptor internalization using the LYN BRET assay as described in Section 2. Schematic diagrams illustrating each assay are depicted above each panel. In each panel, the data are expressed as a percentage of the maximum dopamine response observed in cells transfected with pcDNA control and are displayed as mean ± SEM values from at least three independent experiments. Average curve parameters (EC_50_ and Emax values) and associated statistics are shown in Table 1.

**Figure 2 biomolecules-13-01552-f002:**
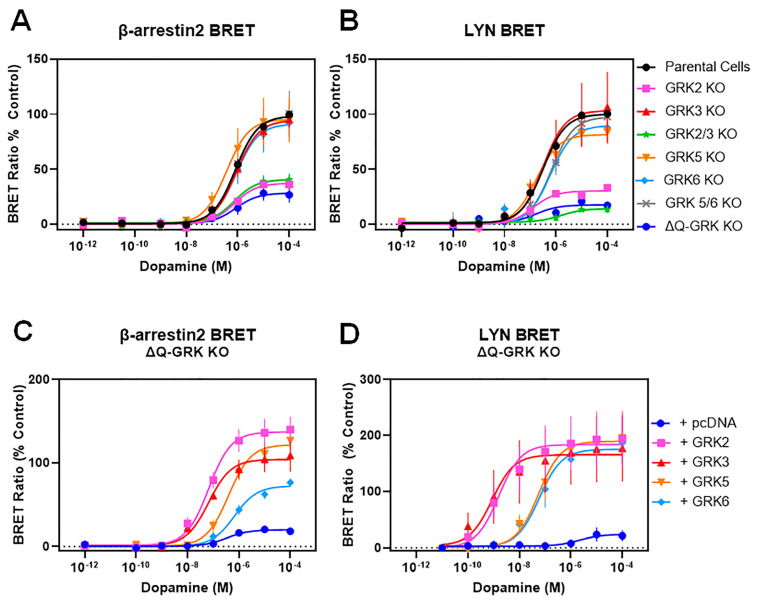
Selective knockout of GRK isoforms reveals the critical role of GRK2 in modulating β-arrestin2 recruitment to the D2R as well as receptor internalization. β-arrestin2 BRET and LYN BRET assays were performed as described in Figure 1 and Section 2 using parental HEK293 cells or HEK293 cells in which the indicated GRK isoform(s) was/were deleted (KO) using CRISPR/Cas9 technology as described in Drube et al., 2022 [28] (Note that ΔQ-GRK KO refers to the simultaneous deletion of GRK2, GRK3, GRK5, and GRK6). (**A**) Concentration-response curves for dopamine stimulation of β-arrestin2-mVenus recruitment to D2R-Rluc8 using the β-arrestin2 BRET assay were generated as described in Figure 1 and Section 2. (**B**) Concentration-response curves for dopamine-stimulated receptor internalization using the LYN BRET assay were performed as described in Figure 1 and Section 2. (**C**) Plasmids expressing either GRK2, GRK3, GRK5, GRK6, or pcDNA (control) were transiently transfected into the ΔQ-GRK KO cell line and dopamine-stimulated β-arrestin2 recruitment was measured as described in panel (**A**). (**D**) Plasmids expressing either GRK2, GRK3, GRK5, GRK6, or pcDNA (control) were transiently transfected into the ΔQ-GRK KO cell line and dopamine-stimulated D2R internalization was measured as described in panel (**B**). Average curve parameters (EC_50_ and Emax values) and statistical comparisons are shown in Table 2 (panels (**A**,**B**)) and Table 3 (panels (**C**,**D**)). In all experiments, the data are expressed as a percentage of the maximum dopamine response observed in the parental HEK293 cells (data shown in Table 3) and are displayed as mean ± SEM values from at least three experiments.

**Figure 3 biomolecules-13-01552-f003:**
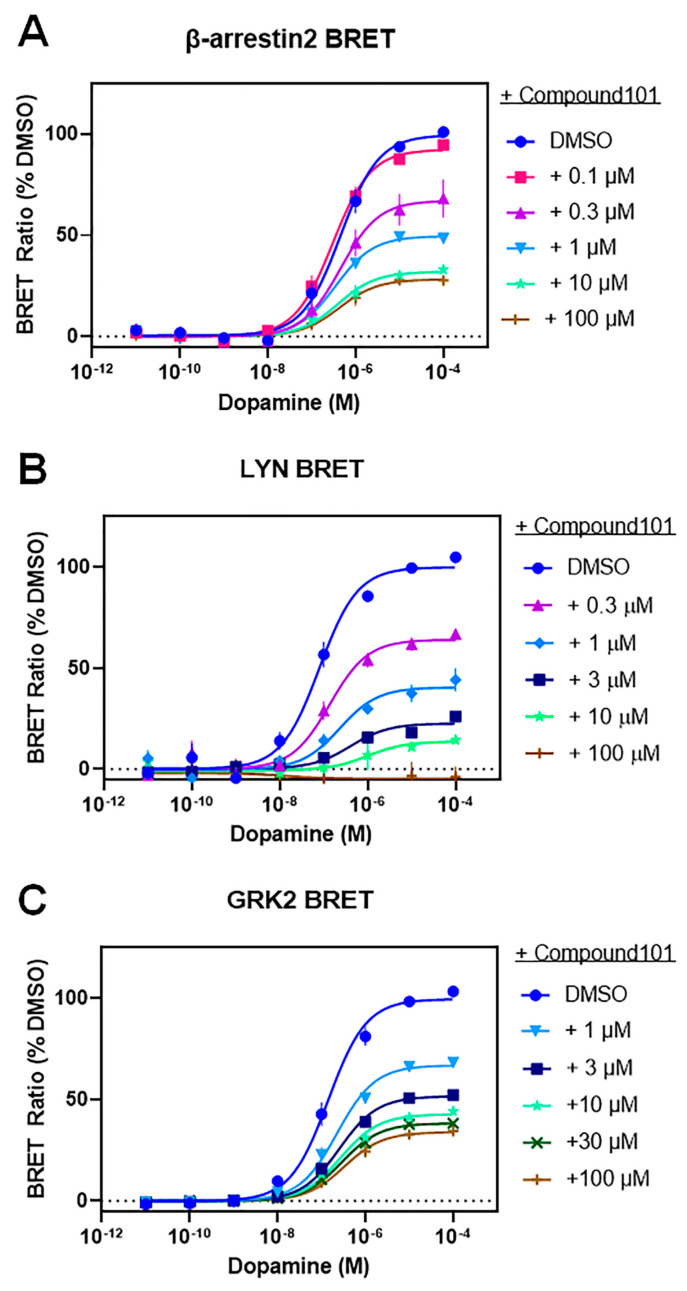
Compound 101 dose-dependently inhibits dopamine-stimulated β-arrestin2 recruitment and receptor internalization as well as D2R–GRK2 interactions in HEK293 cells. Dopamine concentration-response curves were generated in the absence or presence of increasing concentrations of compound 101, as indicated. (**A**) β-arrestin2 recruitment BRET assays were generated as described in Figure 1 and Section 2. (**B**) D2R internalization was assessed using the LYN BRET assay as described in Figure 1 and Section 2. (**C**) D2R–GRK2 interactions were assessed using the GRK2 BRET assay as described in Section 2. Data are expressed as a percentage of the maximum dopamine-stimulated response in cells treated with DMSO and are displayed as mean ± SEM values from at least three experiments. Average curve parameters (EC_50_ and Emax values) and statistical comparisons are shown in Table 4.

**Figure 5 biomolecules-13-01552-f005:**
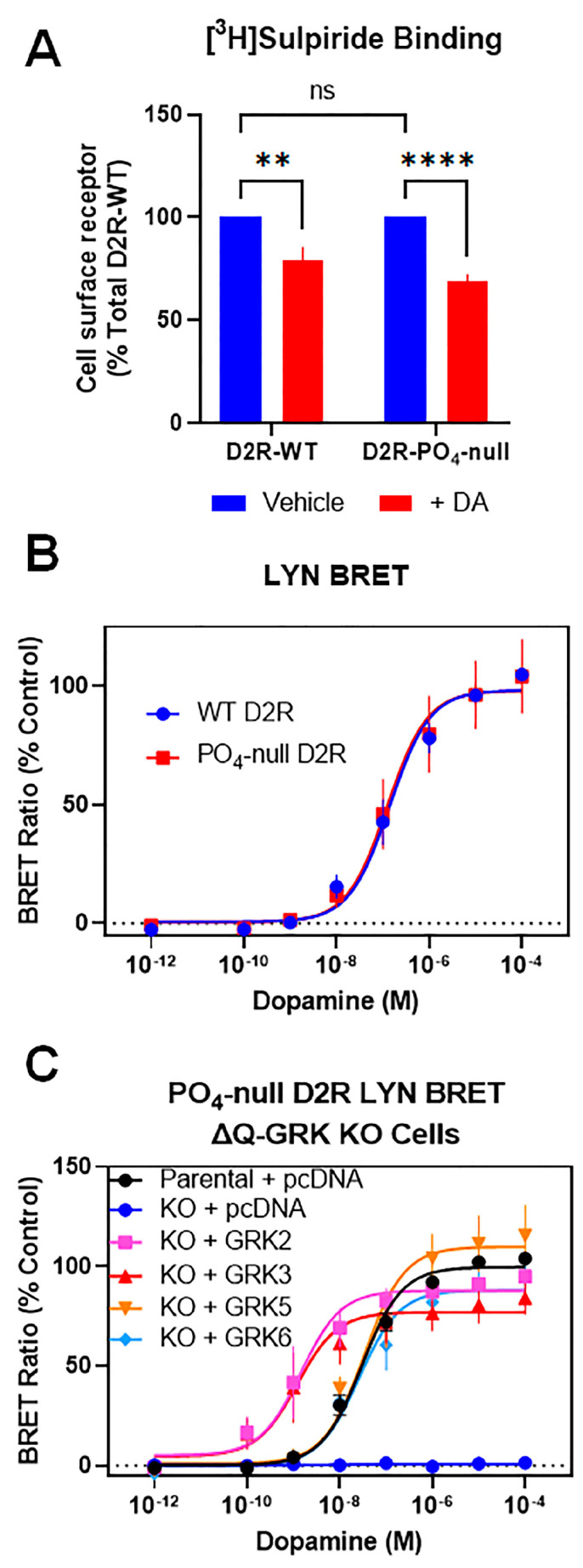
GRKs enhance D2R internalization in the absence of receptor phosphorylation. (**A**) HEK293 cells transiently expressing GRK2 and either WT D2R or PO_4_-null D2R were incubated for 1.5 h with either vehicle or 10 µM dopamine followed by intact cell surface receptor binding using [^3^H]-sulpiride as described in Section 2. Data are expressed as a percentage of the maximum D2R binding for WT D2R in the absence of dopamine. Data points represent mean ± SEM values from at least 5 independent experiments. Statistical significance of the differences between vehicle and dopamine-treated conditions were calculated using two-way ANOVA followed by Tukey’s multiple comparisons test: ** *p* < 0.005, **** *p* < 0.0001, ns = not significant. (**B**) HEK293 cells were transiently transfected with either WT-D2R or PO_4_-null D2R, and LYN BRET D2R internalization assays were performed as described in Section 2. Data are expressed as a percentage of the maximum response (Emax) observed with WT D2R and are shown as mean ± SEM values of at least three individual experiments performed in triplicate. WT D2R: EC_50_ = 250 ± 120 nM, E_max_ = 100%; PO_4_-null D2R: EC_50_ = 260 ± 110 nM, E_max_ = 100 ± 14%. The WT and PO_4_-null D2R curve parameters were compared statistically using a t test and were found not to differ (*p* > 0.05). (**C**) The PO_4_-null D2R and LYN-rGFP were transfected into either parental HEK293 cells or the ΔQ-GRK KO cells. The ΔQ-GRK KO cells were also transfected with either GRK2, GRK3, GRK5, GRK6, or pcDNA (control). The data are expressed as a percentage of the maximum dopamine response observed in parental HEK293 cells and are displayed as mean ± SEM values from at least three experiments. Average curve parameters (EC_50_ and Emax values) and statistical comparisons are shown in Table 5.

**Figure 6 biomolecules-13-01552-f006:**
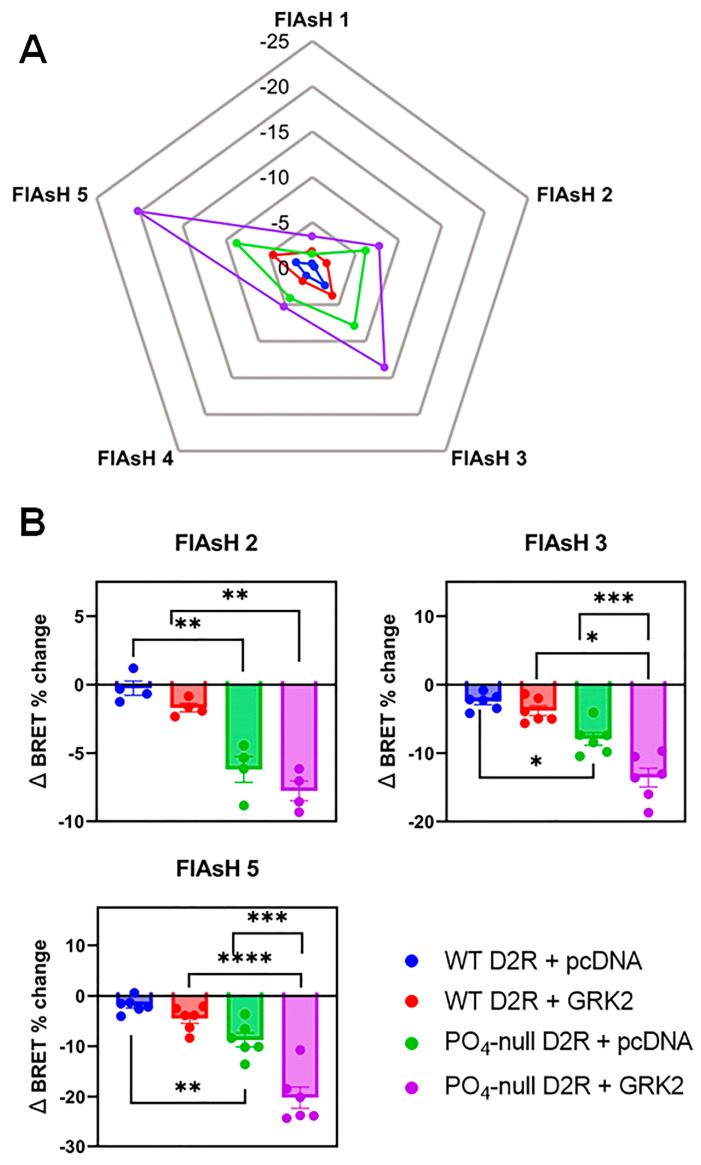
Dopamine-stimulated β-arrestin2 conformational changes are enhanced by GRK2 in the absence of receptor phosphorylation. HEK293 cells were transiently transfected with the indicated D2R construct, the indicated Nluc-β-arrestin2-FlAsH biosensor, and either pcDNA or the GRK2 construct as described in Section 2. These FlAsH biosensors [29] were used as described in Section 2 to detect conformational changes in β-arrestin2 following incubation with either vehicle or 100 µM dopamine for 15 min. The data are expressed as the Δ BRET % change. First, dopamine-stimulated net BRET was calculated by subtracting the BRET ratio of vehicle-treated cells from the BRET ratio of dopamine-treated cells. This value was then divided by the BRET ratio of vehicle-treated cells and then multiplied by 100 to achieve the Δ BRET % change. (**A**) Mean data from at least three individual experiments of each β-arrestin2 FlAsH biosensor studied are displayed in a radar plot. (**B**) Results for selected β-arrestin2 FlAsH biosensors are shown. Data points represent mean ± SEM from each individual experiment. Statistical comparisons were performed using a two-way ANOVA with Tukey’s multiple comparison test: * *p* < 0.05, ** *p* < 0.005, *** *p* < 0.001, **** *p* < 0.0001.

**Figure 7 biomolecules-13-01552-f007:**
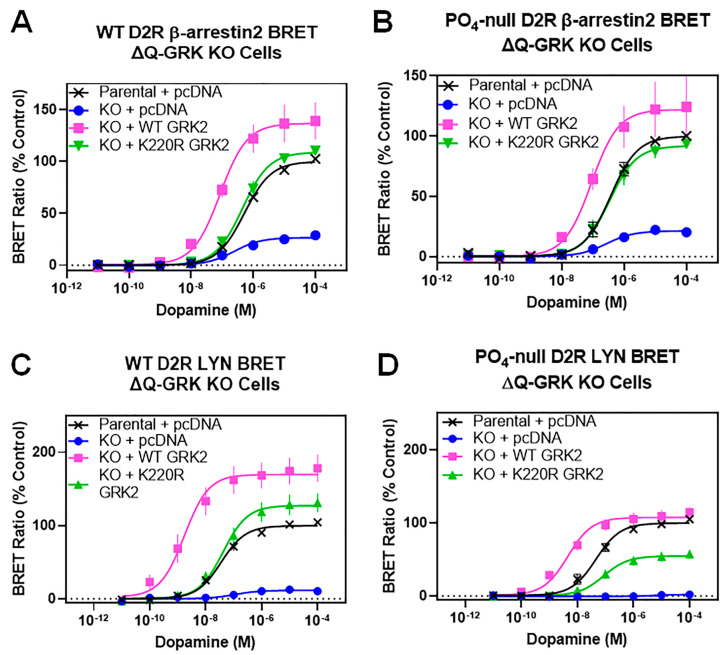
The catalytically inactive K220R GRK2 mutant promotes β-arrestin2 recruitment and D2R internalization in ΔQ-GRK KO cells. β-arrestin2 BRET and LYN BRET assays were performed as described in Figure 2C,D and Section 2, using either parental HEK293 cells or ΔQ-GRK KO cells. (**A**) Plasmids expressing the WT D2R-Rluc8 and either the WT GRK2, K220R GRK2, or pcDNA (control) were transiently transfected into the ΔQ-GRK KO cell line and dopamine-stimulated β-arrestin2 recruitment was measured using a 20 min incubation time. (**B**) Plasmids expressing the PO_4_-null D2R-Rluc8 and either the WT GRK2, K220R GRK2, or pcDNA (control) were transiently transfected into the ΔQ-GRK KO cell line and dopamine-stimulated β-arrestin2 recruitment was measured using a 20 min incubation time. (**C**) Plasmids expressing the WT D2R-Rluc8 and either the WT GRK2, K220R GRK2, or pcDNA (control) were transiently transfected into the ΔQ-GRK KO cell line and dopamine-stimulated D2R internalization was measured using the LYN BRET assay. (**D**) Plasmids expressing the PO_4_-null D2R-Rluc8 and either the WT GRK2, K220R GRK2, or pcDNA (control) were transiently transfected into the ΔQ-GRK KO cell line and dopamine-stimulated D2R internalization was measured using the LYN BRET assay. In all panels, data are expressed as a percentage of the maximum dopamine response observed in the parental HEK293 cells and are displayed as mean ± SEM values from at least three independent experiments. Average curve parameters (EC_50_ and Emax values) and statistical comparisons are shown in Table 7.

**Table 1 biomolecules-13-01552-t001:** GRK isoforms have differential effects on β-arrestin2 recruitment to the D2R and on receptor internalization. Curve parameters are derived from Figure 1. EC_50_ and Emax values represent the mean ± SEM values from at least three independent experiments.

Dopamine-Stimulated		+pcDNA	+GRK2	+GRK3	+GRK5	+GRK6
β-arrestin2 BRET	EC_50_ (nM)	190 ± 28	29 ± 3.3 ****	24 ± 4.8 ****	190 ± 19	230 ± 23
	Emax (% + pcDNA)	100	150 ± 12 *	160 ± 5.1 *	240 ± 20 ****	150 ± 16 *
LYN BRET	EC_50_ (nM)	90 ± 24	1.6 ± 0.5 ****	2.1 ± 1.0 ****	36 ± 11	43 ± 14
	Emax (% + pcDNA)	100	190 ± 19 **	150 ± 15 *	190 ± 23 **	180 ± 7.8 **

Statistical comparisons between the + pcDNA control curve parameters (pEC_50_ values were used for statistical analyses) and the + GRK experimental groups were made using a one-way ANOVA with Dunnett’s multiple comparisons test: * *p* < 0.05, ** *p* < 0.01, **** *p* < 0.0001.

**Table 2 biomolecules-13-01552-t002:** Selective knockout of GRK isoforms reveals the critical role of GRK2 in modulating both β-arrestin2 recruitment to the D2R and receptor internalization. Curve parameters are derived from Figure 2A,B. EC_50_ and Emax values represent the mean ± SEM values from at least three independent experiments.

Dopamine-Stimulated		Parental HEK293	GRK2 KO	GRK3 KO	GRK2/3 KO	GRK5 KO	GRK6 KO	GRK 5/6 KO	ΔQ-GRK KO
β-arrestin2 BRET	EC_50_ (nM)	1300 ± 340	1000 ± 340	1200 ± 410	810 ± 270	580 ± 120	1600 ± 530	1400 ± 490	1600 ± 570
	Emax (% Parental)	100	38 ± 2.9 ***	96 ± 7.7	41 ± 5.6 ***	96 ± 21	92 ± 17	99 ± 20	29 ± 6.9 ****
LYN BRET	EC_50_ (nM)	380 ± 130	160 ± 38	410 ± 160	ND	205 ± 76	660 ± 110	800 ± 280	ND
	Emax (% Parental)	100	31 ± 0.7 ***	104 ± 30	ND	82 ± 11	90 ± 5.7	98 ± 9.3	ND

Statistical comparisons between the HEK293 parental curve parameters (pEC_50_ values were used for statistical analyses) and the GRK KO experimental groups were made using a one-way ANOVA with Dunnett’s multiple comparisons test: *** *p* < 0 0.001, **** *p* < 0.0001. ND = not detectable.

**Table 3 biomolecules-13-01552-t003:** Reintroduction of GRK isoforms into ΔQ-GRK KO cells rescues β-arrestin2 recruitment to the D2R and receptor internalization. Curve parameters are derived from Figure 2C,D. EC_50_ and Emax values represent the means ± SEM from at least three independent experiments.

Dopamine- Stimulated		Parental + pcDNA	ΔQ-KO + pcDNA	ΔQ-KO + GRK2	ΔQ-KO + GRK3	ΔQ-KO + GRK5	ΔQ-KO + GRK6
β-arrestin2 BRET	EC_50_ (nM)	500 ± 110	390 ± 150	81 ± 37 *	71 ± 31 **	460 ± 150	890 ± 410
	Emax (% Parental)	100	20 ± 2.2 ***	140 ± 16	105 ± 17	120 ± 16	73 ± 4.7
LYN BRET	EC_50_ (nM)	91 ± 37	ND	2.6 ± 1.0 ***	3.9 ± 2.1 ***	81 ± 32	110 ± 67
	Emax (% Parental)	100	ND	180 ± 47	170 ± 55	190 ± 42	180 ± 35

Statistical comparisons between the HEK293 parental curve parameters (pEC_50_ values were used for statistical analyses) and the ΔQ-GRK KO cell + GRK experimental groups were made using a one-way ANOVA with Dunnett’s multiple comparisons test: * *p* < 0.05, ** *p* < 0.01, *** *p* <0 0.001, ND = not detectable.

**Table 4 biomolecules-13-01552-t004:** Compound 101 (C-101) inhibits dopamine-stimulated β-arrestin2 recruitment and receptor internalization as well as D2R–GRK2 interactions in HEK293 cells. Curve parameters are derived from Figure 3. Data are expressed as a percentage of the maximum dopamine-stimulated response in cells treated with DMSO and are displayed as mean ± SEM values from at least three independent experiments.

Dopamine-Stimulated		DMSO	+C-101 0.1 µM	+C-101 0.3 µM	+C-101 1 µM	+C-101 3 µM	+C-101 10 µM	+C-101 30 µM	+C-101 100 µM
β-arrestin2 BRET	EC_50_ (nM)	560 ± 170	330 ± 81	470 ± 59	380 ± 130	NT	660 ± 260	NT	570 ± 280
	Emax (% DMSO)	100	92 ± 1.2	67 ± 8.4 ****	50 ± 2.0 ****	NT	33 ± 1.2 ****	NT	28 ± 1.7 ****
LYN BRET	EC_50_ (nM)	85 ± 25	NT	160 ± 49	300 ± 120	580 ± 360	250 ± 130	NT	ND
	Emax (% DMSO)	100	NT	64 ± 2.6 **	41 ± 4.6 ****	23 ± 0.8 ****	23 ± 9.5 ****	NT	ND
GRK2 BRET	EC_50_ (nM)	170 ± 56	NT	NT	260 ± 98	280 ± 76	300 ± 85	300 ± 47	370 ± 88
	Emax (% DMSO)	100	NT	NT	67 ± 2.4 ****	52 ± 1.1 ****	43 ± 1.3 ****	38 ± 2.4 ****	34.0 ± 2.2 ****

Statistical comparisons between the DMSO control curve parameters (pEC_50_ values were used for statistical analyses) and the compound 101 treatment groups were made using a one-way ANOVA with Dunnett’s multiple comparisons test: ** *p* < 0.005, **** *p* < 0.0001. NT = not tested, ND = not detectable.

**Table 5 biomolecules-13-01552-t005:** Receptor phosphorylation is not required for GRK2-mediated enhancement of dopamine-stimulated recruitment of β-arrestin2 to the D2R or receptor internalization. Curve parameters are derived from Figure 4C (β-arrestin2 BRET) and Figure 5C (LYN BRET). Data are expressed as a percentage of the maximum dopamine-stimulated response in the parental HEK293 cells and are displayed as mean ± SEM values from at least three independent experiments.

Dopamine- Stimulated		Parental + pcDNA	ΔQ-KO + pcDNA	ΔQ-KO + GRK2	ΔQ-KO + GRK3	ΔQ-KO + GRK5	ΔQ-KO + GRK6
β-arrestin2 BRET	EC_50_ (nM)	99 ± 29	106 ± 21	20 ± 6.5 ***	28 ± 7.7 *	160 ± 39	120 ± 8.6
	Emax (% Parental)	100	22 ± 1.1 ****	81 ± 13	85 ± 8.3	113 ± 12	102 ± 23
LYN BRET	EC_50_ (nM)	34 ± 8.0	ND	2.5 ± 2.0 *	2.6 ± 2.2 *	53 ± 38	49 ± 33
	Emax (% Parental)	100	ND	88 ± 3.5	77 ± 7.7	103 ± 13	90 ± 8.5

Statistical differences between the curve parameters for the parental HEK293 + pcDNA and other experimental conditions were performed using a one-way ANOVA with Dunnett’s multiple comparisons test. pEC_50_ values were used for the statistical analyses: * *p* < 0.05; *** *p* < 0.001; **** *p* < 0.0001. ND = not detectable.

**Table 6 biomolecules-13-01552-t006:** Compound 101 (C-101) inhibits dopamine-stimulated β-arrestin2 recruitment to the PO_4_-null D2R in HEK293 cells. Curve parameters are derived from Figure 4D. Data are expressed as a percentage of the maximum dopamine-stimulated response in cells treated with DMSO and are displayed as mean ± SEM values from at least three independent experiments.

Dopamine-Stimulated		DMSO	+C-101 0.1 µM	+C-101 0.3 µM	+C-101 1 µM	+C-101 10 µM	+C-101 100 µM
β-arrestin2 BRET	EC_50_ (nM)	410 ± 120	250 ± 21	440 ± 140	280 ± 46	430 ± 140	580 ± 230
	Emax (% DMSO)	100	92 ± 3.1	68 ± 5.0 ****	49 ± 1.6 ****	33 ± 1.9 ****	26 ± 4.6 ****

Statistical comparisons between the DMSO control curve parameters (pEC_50_ values were used for statistical analyses) and the compound 101 treatment groups were made using a one-way ANOVA with Dunnett’s multiple comparisons test: **** *p* < 0 0.0001.

**Table 7 biomolecules-13-01552-t007:** The catalytically inactive K220R GRK2 mutant promotes β-arrestin2 recruitment and D2R internalization in ΔQ-GRK KO cells. Curve parameters are derived from Figure 7. Data are displayed as mean ± SEM values from at least three independent experiments.

Dopamine- Stimulated			Parental + pcDNA	ΔQ-KO + pcDNA	ΔQ-KO + WT GRK2	ΔQ-KO + K220R GRK2
β-arrestin2 BRET	WT D2R	EC_50_ (nM)	530 ± 78	370 ± 210	91 ± 34 *	480 ± 130
		Emax (% Parental)	100	23 ± 4.1 ****	135 ± 13 *	101 ± 8.6
	PO_4_-null D2R	EC_50_ (nM)	400 ± 130	270 ± 14	88 ± 22 *	400 ± 140
		Emax (% Parental)	100	21 ± 0.8 **	122 ± 24	92 ± 3.1
LYN BRET	WT D2R	EC_50_ (nM)	41 ± 10	ND	2.4 ± 0.8 **	48 ± 13
		Emax (% Parental)	100	ND	170 ± 18 **	130 ± 13
	PO_4_-null D2R	EC_50_ (nM)	53 ± 13	ND	4.9 ± 1.3 **	110 ± 25
		Emax (% Parental)	100	ND	107 ± 8.3	55 ± 6.0 ***

Statistical differences between the curve parameters for the parental HEK293 + pcDNA group and the ΔQ-KO + pcDNA or GRK experimental groups were performed using a one-way ANOVA with Dunnett’s multiple comparisons test. (pEC_50_ values were used for the statistical analyses): * *p* < 0.05; ** *p* < 0.01; *** *p* < 0.001; **** *p* < 0.0001. ND = not detectable.

## Data Availability

The data presented in this study are available on request from the corresponding author.

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
