# Peer review of "G Protein-Coupled Receptor Kinase 2 Selectively Enhances β-Arrestin Recruitment to the D2 Dopamine Receptor through Mechanisms That Are Independent of Receptor Phosphorylation"

_biomolecules, 2023, doi:10.3390/biom13101552_

Round 1
Reviewer 1 Report
It is believed that β-arrestins interact with G protein-coupled receptors (GPCRs) at the phosphorylated C-terminal tail or intracellular loops. GPCR kinases (GRKs), which are responsible for GPCR phosphorylation, are indispensable for β-arrestin recruitment but not for D2R-β-arrestin interactions. The manuscript aims to explore the role of GRKs in D2R-β-arrestin interactions. After experiments with GRK knockout cells, it was found that genetic elimination of all GRK expression decreased, but did not eliminate, agonist-stimulated β-arrestin recruitment to the D2R or its subsequent internalization. Next, it was found that these processes were rescued upon reintroduction of various GRK isoforms in the cells with GRK2/3 also enhancing dopamine potency. Treatment with compound 101, a pharmacological inhibitor of GRK2/3 isoforms, decreased β-arrestin recruitment and receptor internalization, highlighting the importance of this GRK subfamily for D2R-β-arrestin interactions. These results were recapitulated using a phosphorylation-deficient D2R mutant. emphasizing that GRKs can enhance β-arrestin recruitment and activation independently from receptor phosphorylation. The manuscript is well-structured, and the study was designed appropriately.
However, the Reviewer found some concerns regarding the presentation of the results. Some parts remind more of discussion than results presentation.
Minor: page 13, 455-457: “The reason(s) for these discrepant results is not immediately clear but might be due to differences in the physical orientation of the BRET biosensors incorporated into the D2R and GRK2.”
Could you explain this?
Reviewer 2 Report
The manuscript by Sanchez-Soto et al. describes the specificity of D2 dopamine receptor (D2R) and the G-protein coupled receptor kinase-2 (GRK2) interaction in beta-arrestin recruitment and receptor internalization. BRET assays were done for the roles of all possible GRK isoforms and establishing genetic GRK KO cells they found that each of them has positive impacts on dopamine stimulated arrestin recruitment and internalization, however, only GRK2/3 are able to enhance dopamine potency. The authors have done further experiments on their previous finding that the effect of GRKs may have been independent of D2R phosphorylation and this was confirmed using WT and phosphorylation deficient mutants of D2R. In addition, compound 101, a pharmacological inhibitor of GRK2/3 catalytic activity, attenuated beta-arrestin recruitment and receptor internalization. The authors conclude that two different GRK-mediated mechanisms exist for enhancing β-arrestin recruitment to the D2R, a phosphorylation independent one, and the other may be dependent on phosphorylation by GRK2/3, but the physiological substrate(s) is/are yet to be identified. The authors used reliable methotologies to prove their findings providing also a vast amount of supplementary data. The manuscript is well organized, the data are straightforward and it is written in a compact way. It certainly merits publication in this journal. No major concerns are raised, however, some minor notes are presented below.
11. The Discussioon of the manuscript is too long and it is not concise enough. In the first part, it mainly repeats the results without any comparision with previous literatura data or presenting any hypothesis on their data.
22. There are some discrepancies of the results obtained with the GRK2/3 specific inhibitor and the catalytically inactive mutant GRK2. While the both seems to affect mainly the domamine potency, the inhibitor suppress also the response, too. I think it is not elaborated in sufficient details in the manuscript. The described effects of the inhibitor appears to be an enigma regarding its lack of impact on the phospho-null-mutant, but the authors fail to give any considerable biochemical background to this effect.
Reviewer 3 Report
This is an interesting article in which the authors investigate the mechanisms of GRK-mediated enhancement of arrestin recruitment to and internalization of the dopamine D2 receptor. The paper should be of considerable interest to the field. However, some issues should be addressed before the paper can be accepted for publication.
Major issues
It would be important to clarify which receptor isoform (D2L or D2S) was used in this study, especially since isoform-specific properties of D2 receptor desensitization and its regulation by GRK2 have been reported (Gantz et al., Elife, 2015, PMID: 26308580; Ågren and Sahlholm, 2021, FASEB J, PMID: 34699610). It would also be helpful to state the amounts of co-transfected constructs (receptors, GRKs, arrestins) in the figures or their legends.
Since the authors mention that endogenous GRK2 expression is higher than that of GRK3, could the stronger effect of knocking out GRK2, as opposed to GRK3, simply relate to these differences in endogenous expression levels? This possibility is hinted at in the discussion, but could be stated more clearly if this is what the authors believe.
The authors report that coexpression of GRKs, in particular GRK2, enchances arrestin recruitment to the dopamine D2 receptor even in the absence of receptor phosphorylation by GRK2. The enhancement of D2 arrestin-dependent desensitization by coexpressed GRK2 in the presence of Cmpd101 or using a K220R mutant GRK2 was recently reported by Agren and Sahlholm (2021, FASEB J, PMID: 34699610). As mentioned above, the effect of Cmpd101 in this study was dependent on the D2 isoform studied. This previous study should be cited and discussed in the context of the present findings.
Finally, the authors should present the results of the phospho-proteomic studies, even if inconclusive, that are mentioned in the Discussion. At least, these results could be included as supplementary material.
Minor issues
line 828 - remove first "However"
